# Application of Dynamic Beam Positioning for Creating Specified Structures and Properties of Welded Joints in Electron-Beam Welding

**DOI:** 10.3390/ma13102233

**Published:** 2020-05-13

**Authors:** Tatyana Olshanskaya, Vladimir Belenkiy, Elena Fedoseeva, Elena Koleva, Dmitriy Trushnikov

**Affiliations:** 1Department of Welding Production, Metrology and Material Technology, Perm National Research Polytechnic University, Perm 614990, Russia; tvo66@mail.ru (T.O.); vladimirbelenkij@yandex.ru (V.B.); elena.fedoseeva.79@mail.ru (E.F.); 2Faculty of Physics, University of Chemical Technology and Metallurgy, 1756 Sofia, Bulgaria; eligeorg@abv.bg

**Keywords:** electron-beam welding, welded metal structure, dynamic positioning of an electron beam, electron beam

## Abstract

The application of electron beam sweep makes it possible to carry out multifocal and multi-beam welding, as well as combine the welding process with local heating or subsequent heat treatment, which is important when preparing products from thermally-hardened materials. This paper presents a method of electron beam welding (EBW) with dynamic beam positioning and its experimental-calculation results regarding the formation of structures and properties of heat-resistant steel welded joints (grade of steel 20Cr3MoWV). The application of electron beam oscillations in welding makes it possible to change the shape and dimensions of welding pool. It also affects the crystallization and formation of a primary structure. It has been established that EBW with dynamic beam positioning increases the weld metal residence time and the thermal effect zone above the critical *A*_3_ point, increases cooling time and considerably reduces instantaneous cooling rates as compared to welding without beam sweep. Also, the difference between cooling rates in the depth of a welded joint considerably reduces the degree of structural non-uniformity. A bainitic–martensitic structure is formed in the weld metal and the thermal effect zone throughout the whole depth of fusion. As a result of this structure, the level of mechanical properties of a welded joint produced from EBW with dynamic electron beam positioning approaches that of parent metal to a greater extent than in the case of welding by a static beam. As a consequence, welding of heat-resistant steels reduces the degree of non-uniformity of mechanical properties in the depth of welded joints, as well as decreases the level of hardening of a welded joint in relation to parent metal.

## 1. Introduction

Electron beam welding (EBW) is being increasingly implemented in various applications [1], including the manufacturing of essential products. In some cases, EBW is used at the final stage of manufacturing the products from thermally-hardened materials with a given set of properties, which makes it difficult or impossible to carry out subsequent heat treatment of welded joints. At the same time, welded joint properties must be similar to those of the parent product material [2]. It is important to note that EBW can have the desired effect not only on solid and high-melting alloys but also on thermally hardened materials. High-quality joints are possible from the local effects of temperature [3,4]. The level of hardening of welded joints from heat-resistant steels in products for which there is only a low tempering rate is high enough. It is commonly known that the physical weldability of heat-resistant steels is hampered by the tendency of welded joints to develop cold cracks and soften metals in the thermal effect zone (TEZ). Therefore, products should be locally or preliminarily heated as part of the welding process. This decreases the temperature difference in the welding zone and peripheral areas, reducing metal stresses. The metal cooling rate decreases, and after welding, an increasingly large amount of austenite transforms to martensite at high temperature when the metal is flexible. Stresses occurring due to the difference in the volumes of these phases will decrease, and cold cracks will be less probable. It should be taken into account during heating that abnormally high temperatures result in the formation of coarse ferrite–perlite structures that do not ensure required long-term strength and impact toughness of welded joints. It is possible to reduce the risk of cold cracking by tempering parts at the temperature of 150–200 °C immediately after welding for several hours. During this time, the transformation of the residual austenite to martensite will be completed, and most of the hydrogen dissolved in it will be removed from the metal.

The softening of heat-resistant steels in TEZ also depends on the temperature-time parameters of welding. An increase in heat input enlarges a soft interlayer in TEZ, which may disrupt rigid weld joints in the course of operation, especially under bending loads. Also, structural phase transformations taking place in the welding zone mainly depend on the temperature–time parameters: degree of heating, heat distribution, heating, and cooling rates [5,6,7,8,9].

Due to the high concentration of energy in the electron beam affected zone, EBW is characterized by significant heating and cooling rates, as well as by high temperature gradient values. This causes a considerable non-uniformity of the temperature field that, in turn, leads to chemical, structural, and mechanical non-uniformity of a welded joint [3,10,11].

It is possible to regulate thermal cycles in EBW in order to produce a specified structure of a welded joint though controlling the thermal power of an electron beam due to its dynamic positioning [5]. The application of beam sweep makes it possible to carry out multifocal and multi-beam welding when welding is performed simultaneously in several treated areas [6,7], as well as combine the welding process with local heating or subsequent heat treatment.

When developing an EBW technology for structures from thermally hardened materials, there is the problem of choosing energy parameters of a welding mode. An additional problem is choosing the type and parameters of EBW to reproduce a specified thermal cycle, ensuring the formation of structures and properties of welded joints [10] close to those of parent material.

By using the example of aluminum alloy, we have established the main regularities of the effect of three-bath EBW parameters (beam current, welding speed, relative pulse duration, distance between points) on geometric characteristics of welds (weld depth, width, fusion shape, and fullness coefficients) and obtained regression dependencies. The criteria characterizing the formation of defect-free welds have been determined. The method for determining optimal modes of three-bath EBW for aluminum alloys by drawing nomograms and solving a system of equations from regression dependencies [5,7,12] has been proposed. Also, the effect of dynamic deflection of an electron beam on the structure and properties of welded joints from heterogeneous materials has been studied using the example of steel-bronze [9,13,14]. Based on the studies, we have developed technological recommendations to reduce structural and mechanical non-uniformity of welded joints from steel and bronze.

To the best of our knowledge, there are no studies on the effect of dynamic beam deflection in welding on the structure and properties of welded joints from heat-hardenable medium-alloy steels. However, EBW with dynamic beam positioning aimed at producing equal-strength heat-resistant steel welded joints and improving their quality continues to be of particular relevance and importance.

The purpose of this work was to study and develop a technology aimed at reducing the level of hardening of welded joints of heat-resistant steels in products for which only low-temperature tempering was provided. We present the results of studies on the effect of EBW with dynamic beam positioning on the formation of structures and properties of heat-resistant steel welded joints. The studies were conducted to develop a technology aimed at reducing the level of hardening of heat-resistant steel welded joints in products for which only low-temperature tempering is provided. The EBW process with a sweep along the trajectory ensuring concurrent heating of welded edges was studied. The application of electron beam oscillations in welding makes it possible to change welding pool shape and dimensions, as well as affect the crystallization and formation of a primary structure. The use of dynamic positioning of an electron beam with heating of welded joint edges in EBW increases the width of a fusion zone, which affects the crystallization process and decreases cooling rates and their range of change in depth. This reduces the degree of non-uniformity of macro- and microstructure in welded joints.

## 2. Materials and Methods

This work includes experimental and calculation studies on the effect of dynamic positioning of an electron beam in EBW on the structure and properties of welded joints from the 20Cr3MoWV heat-resistant steel. The chemical composition of this grade of steel is given in Table 1.

A comparative analysis of the resulting structure and properties of a welded joint was conducted in EBW without beam oscillations, with X-shaped oscillations and with combined beam oscillations along the trajectory simulating multi-beam welding. When forming this trajectory simulating multi-beam welding, an electron beam on the metal surface was dynamically positioned such that it affected three zones, thereby forming two parallel lines on both sides of the joint at some distance from it and a point located on the joint line. The trajectory of combined electron beam oscillation is presented in Figure 1.

For the selection of an electron beam positioning trajectory and its parameters, it was considered that the number of points along which the electron beam moves increases, while the depth of fusion decreases. In this case, the purpose was to obtain a fusion depth of up to 10 mm. To increase the length of a welding pool, beam oscillations along the joint were selected, and to increase its width, longitudinal oscillations were shifted in relation to the joint. The third electron beam exposure zone, a point located on the joint line, ensured the predetermined depth of fusion. The distance between the electron beam exposure zones was to have one general fusion channel formed by the electron beam in the metal (the size of the oscillation path must be small enough so that the area of the beam impact is limited to one common penetration channel). The parameters of this trajectory of dynamic positioning of an electron beam were determined based on preliminary calculations using thermal models [5,6,9]. The following parameters were determined: electron beam power—4000 W, welding speed—5 mm/s, electron beam operation time in each zone—250 μs, and longitudinal oscillation frequency—1000 Hz. The sharp focusing mode was used. The electron beam diameter (the size of an area where 95% of the electron beam energy is concentrated) was measured using the procedure and equipment described previously [15,16] and amounted to 0.35 mm (beam power—2000 W) and 0.5 mm (beam power—4000 W) on the surface of the workpiece.

There are intense convective flows in the weld pool. At the same time, the structure of the material is determined by thermal processes and depends on thermal cycles. In this work, only the thermal problem is solved, and the convective flows are not taken into account.

To implement the described trajectory, the electron gun of the ELA-6VCH power unit by SELMI (Ukraine) was upgraded through installing a high-speed deflecting system, where a signal on coils was sent from the outputs of a two-channel broadband amplifier. The amplifier was connected via a digital-to-analogue interface to a computer with installed software, allowing dynamic positioning of an electron beam along various types of user-defined trajectories. Heat-resistant steel 20Cr3MoWV was chosen as a weld material (Figure 2).

Table 2 shows the EBW parameters for 20Cr3MoWV steel when carrying out the experimental studies.

The calculation studies were conducted based on mathematical models allowing the assessment of the effect of the beam oscillation trajectory and its parameters on changes in a fusion form, thermal welding cycles in the weld metal and the TEZ, instantaneous cooling rates, as well as on the conditions of formation of macro- and microstructure of welded joints. The Mathcad application program package was used for calculations.

A thermal model based on differential equations for thermal conductivity in the mobile coordinate system with a fixed source was used to construct a fusion form and thermal welding cycles. This model was obtained by an analytical method using Green’s functions [1,3,4,5]. A standard integral solution to a thermal conductivity problem in the mobile coordinate system for an endless plate, with different types of dynamic electron beam positioning taken into account, is
(1)$$\begin{array}{c} T\left(x,y,z,t\right)\\ =\overset{\infty }{\underset{-\infty }{\int }}\overset{\infty }{\underset{-\infty }{\int }}\overset{S}{\underset{0}{\int }}\overset{\infty }{\underset{0}{\int }}\frac{1}{8{\left(\sqrt{\pi a\left(t-\tau \right)}\right)}^{3}}exp\left(-\frac{{\left(x-x^{\prime }+V\left(t-\tau \right)\right)}^{2}}{4a\left(t-\tau \right)}\right)exp\left(-\frac{{\left(y-y^{\prime }\right)}^{2}}{4a\left(t-\tau \right)}\right)\\ \cdot \overset{\infty }{\underset{n=-\infty }{\sum }}\left(\begin{array}{c} exp\left(-\frac{{\left(z-z^{\prime }+2nS\right)}^{2}}{4a\left(t-\tau \right)}\right)+\\ +exp\left(-\frac{{\left(z+z^{\prime }+2nS\right)}^{2}}{4a\left(t-\tau \right)}\right) \end{array}\right)\cdot F\left(x^{\prime },y^{\prime },z^{\prime },t\right)\partial x^{\prime }\partial y^{\prime }\partial z^{\prime }\partial \tau \end{array}$$
(2)$$F\left(x^{\prime },y^{\prime },z^{\prime },t\right)=\frac{\eta q}{c\rho }\cdot \delta \left(x-x^{\prime }\right)\cdot \delta \left(y-y^{\prime }\right)\cdot \delta \left(z-z^{\prime }\right)\cdot \delta \left(\tau \right)$$
where *V*—welding speed; *S*—plate thickness; *F*(*x*′,*y*′,*z*′,*t*)—heat source function described using the Dirac delta function; *x*′, *y*′, *z*′—heat source coordinates; *τ*—source operation time; *q*—electron beam power; *η*—efficiency factor; *c*—specific heat capacity; and *ρ*—metal density. Heat source forms and their mathematical expressions *F*(*x*′,*y*′,*z*′,*t*) for different types of electron beam oscillations are presented in previous works [5,14,15].

Typically, a cooling time within the temperature range of 800–200 °C (*t*_8/2_) and a cooling rate within the range of 600–500 °C (*w*_5/6_) determined by a thermal welding cycle were used to analyze the microstructure of a welded joint. However, heating and cooling rates in the course of welding vary with time nonlinearly, so it is proposed to use instantaneous rates. Equations for determining instantaneous heating and cooling rates in the course of welding with different types of dynamic electron beam positioning were derived from the equation for solving thermal problems (1)
(3)$$W\left(x,y,z,\tau \right)=\frac{dT\left(x,y,z,\tau \right)}{dt}\;\;\;п\;\;р\;\;и\;\;\;dt=\frac{dx}{V}\;\;\;\;W\left(x,y,z,\tau \right)=\frac{dT\left(x,y,z,\tau \right)}{dx}V.$$

A mathematical model developed based on the analytical approach presented in the works [14] was used to analyze the formation of a primary macrostructure of weld metal. To ensure the mathematical setting of a model problem, the following assumptions were made: (1) the crystallization front shape represents a surface described by the crystallization isotherm equation without taking into account the sizes of a two-phase liquid–solid zone; (2) crystallites grow in the direction of temperature gradient and, consequently, their growth axes represent orthogonal trajectories to the crystallization front. The model consists of a number of equations: crystallization front, crystallite growth axis trajectories, direction angles of crystallite inclination towards coordinate planes, crystallite growth rates, as well as equations of crystallization scheme criteria and crystallization rate. Taking into account the specific form of fusion penetration in EBW, the crystallization front equation represents two systems of equations (separately for the upper and lower parts of the weld)
(4)$$\begin{array}{cc} \{\begin{array}{c} {\left(\frac{z}{H_{1}}\right)}^{\omega 1}+{\left(\frac{y}{P_{1}}\right)}^{\vartheta 1}=1\\ {\left(\frac{y}{P_{1}}\right)}^{\eta 1}+{\left(\frac{x}{L_{1}}\right)}^{\nu 1}=1\\ {\left(\frac{z}{H_{1}}\right)}^{\tau 1}+{\left(\frac{x}{L_{1}}\right)}^{\mu 1}=1 \end{array} & \{\begin{array}{c} {\left(\frac{z}{H_{2}}\right)}^{\omega 2}+{\left(\frac{y}{P_{2}}\right)}^{\vartheta 2}=1\;\;\;\;\;\;\;YOZ\;plane\\ {\left(\frac{y}{P_{2}}\right)}^{\eta 2}+{\left(\frac{x}{L_{2}}\right)}^{\nu 2}=1\;\;\;\;\;\;\;\;XOY\;plane\\ {\left(\frac{z}{H_{2}}\right)}^{\tau 2}+{\left(\frac{x}{L_{2}}\right)}^{\mu 2}=1\;\;\;\;\;XOZ\;plane \end{array} \end{array}$$

Each system equation describes the isotherm of crystallisation for a relevant coordinate plane, Figure 3.

The coefficients and parameters of the system of crystallization front Equation (3) were determined by approximating the numerical values (*x_i_*,*y_i_*,*z_i_*) of the isothermal crystallization surface obtained in solving the EBW thermal problem (1), with this system of equations. At the same time, coefficients *ω*, *θ*, *η*, *ν*, *τ*, and *µ* may have any non-integer value greater than 1. Equations for calculating the shape of crystallite axes make it possible to construct crystallite axis projections onto coordinate planes and assess a primary macrostructure of weld metal from a qualitative point of view. The values of crystallite growth origin coordinates (*x*_0_,*y*_0_,*z*_0_) are determined to take into account the system Equation (3)
(5)$$\{\begin{array}{cc} y={\left[\frac{H^{\omega }\cdot \left(\theta^{2}-2\theta \right)}{P^{\theta }\cdot \left(\omega^{2}-2\omega \right)}\left(z^{2-\omega }-z_{0}^{2-\omega }\right)+y_{0}^{2-\theta }\right]}^{1/\left(2-\theta \right)} & YOZ\;plane\\ x={\left[\frac{P^{\eta }\cdot \left(\nu^{2}-2\nu \right)}{L^{\nu }\cdot \left(\eta^{2}-2\eta \right)}\left(y^{2-\eta }-y_{0}^{2-\eta }\right)+x_{0}^{2-\nu }\right]}^{1/\left(2-\nu \right)} & XOY\;plane\\ x={\left[\frac{H^{\tau }\cdot \left(\mu^{2}-2\mu \right)}{L^{µ}\cdot \left(\tau^{2}-2\tau \right)}\left(z^{2-\tau }-z_{0}^{2-\tau }\right)+x_{0}^{2-\mu }\right]}^{1/\left(2-\mu \right)} & XOZ\;plane \end{array}\;$$

Changes in the inclination angles of crystallite axes characterize the spatial orientation of crystallites and allow the numerical determination of changes in their growth rate (Figure 4). To analyze the primary macrostructure, the most indicative are the changes in the inclination angles of crystallite axes growing at different depths to the weld axis in the horizontal plane (*α*) and the vertical plane (*γ*).

Changes in the growth rate of crystallites along the weld width in the horizontal (Vα) and vertical (*V**γ*) planes were determined as
(6)$$V\alpha =V{\left[1+\frac{x_{0}{}^{2}\cdot \eta^{2}}{y_{0}{}^{2}\cdot \nu^{2}}Ky^{2\eta -2}{\left(1-Ky^{\eta }\right)}^{\frac{2}{\nu }-2}\right]}^{-\frac{1}{2}},$$
(7)$$V\gamma =V{\left[1+\frac{H^{2\omega }\cdot \theta^{2}}{P^{2\theta }\cdot \omega^{2}}\frac{{\left(Ky\cdot y_{0}\right)}^{2\theta -2}}{z_{0}{}^{2\omega -2}}{\left(1-Ky^{\theta }\right)}^{\frac{2}{\omega }-2}\right]}^{-\frac{1}{2}},$$
where *V*—welding speed, *Ky* = *y/y*_0_—dimensionless coordinate.

The integral criteria of the Kα and Kγ crystallization scheme make it possible to assess resulting macrostructures in EBW. The Kα criterion characterizes the preferred direction of crystallite axes along the weld width in the horizontal plane, and the Kγ criterion characterizes the preferred direction of crystallite axes along the weld width in the vertical plane
(8)$$K\alpha =\int_{0}^{1}arctg\left[\frac{x_{0}\cdot \eta }{y_{0}\cdot \nu }Ky^{\eta -1}{\left(1-Ky^{\eta }\right)}^{-\frac{\nu -1}{\nu }}\right]dKy,$$
(9)$$K\gamma =\int_{0}^{1}arctg\left[\frac{H^{\omega }\cdot \theta }{P^{\theta }\cdot \omega }\frac{{\left(Ky\cdot y_{0}\right)}^{\theta -1}}{z_{0}{}^{\omega -1}}{\left(1-Ky^{\theta }\right)}^{-\frac{\omega -1}{\omega }}\right]dKy.$$

A crystallite growth rate is quantitatively assessed using the two integral criteria of crystallization rate—*KVα* and *KVγ*, characterizing the total value of relative crystallite growth rate along the weld width in the horizontal and vertical planes
(10)$$KV\alpha =\int_{0}^{1}{\left[1+\frac{H^{2\omega }\cdot \theta^{2}}{P^{2\theta }\cdot \omega^{2}}\frac{{\left(Ky\cdot y_{0}\right)}^{2\theta -2}}{z_{0}{}^{2\omega -2}}{\left(1-Ky^{\theta }\right)}^{\frac{2}{\omega }-2}\right]}^{-1/2}dKy,\;$$
(11)$$KV\gamma =\int_{0}^{1}{\left[1+\frac{H^{2\omega }\cdot \theta^{2}}{P^{2\theta }\cdot \omega^{2}}\frac{{\left(Ky\cdot y_{0}\right)}^{2\theta -2}}{z_{0}{}^{2\omega -2}}{\left(1-Ky^{\theta }\right)}^{\frac{2}{\omega }-2}\right]}^{-1/2}dKy.\;$$

The application of integral criteria of crystallization scheme and crystallization rate makes it possible to construct diagrams characterizing the macrostructure of weld metal.

To analyze the emerging microstructure of welded joints, a method was used based on the construction of a series of structural diagrams depending on the cooling rate. When plotting structural diagrams, the regression models (obtained using artificial neural networks) of the transformation of supercooled austenite under continuous cooling were used [17]. These regression equations determine the type of microstructure formed after cooling with four dichotomous variables containing information on the presence of ferrite, perlite, bainite, and martensite in the structure
(12)$$\begin{array}{cc} & X\left(\%\right)=\{\begin{array}{c} 0\;при\;\;W_{X}=0\\ 0\;при\;\;\;U_{X}\leq 0\\ U_{X}\;при\;\;U_{X}>0 \end{array}\\ U_{X}=a_{0}+a_{1}C+ & a_{2}Mn+a_{3}Si+a_{4}Cr+a_{5}Ni+a_{6}Mo+a_{7}V+a_{8}T_{A}+a_{8}W_{\;o\;\;х\;\;л}\\ & +a_{10}CV_{r}{}^{0.25}+a_{11}W_{f}+a_{12}W_{p}+a_{13}W_{b}+a_{14}W_{m} \end{array}$$
where *C*, *Mn*, *Si*, *Cr*, *Ni*, Mo, *V*—weight fractions of alloying elements; *a*_0_, *a*_1_,…, *a*_14_—coefficients obtained by regression analysis; *U_X_*—volume fraction of the structural component; *X*—type of the structural component; *T_A_*—austenitizing temperature, °C; *W_cool_*—cooling rate, °C/min; *W_X_*— dichotomous variables.

Dichotomous variables are designed to determine the probability of specific microstructural components at a given constant cooling rate and austenitizing temperature
(13)$$\begin{array}{cc} & W_{X}=\exp\left(S_{X}\right)/\left(1+\exp\left(S_{X}\right)\right),\\ S_{X}=b_{0X}+b_{1X}C & +b_{2X}Mn+b_{3X}Si+b_{4X}Cr+b_{5X}Ni+b_{6X}Mo+b_{7X}V+b_{8X}Cu\\ & +b_{9X}T_{A}+b_{10X}W_{oхл} \end{array}$$
where *b*_0*X*_, *b*_1*X*_,… *b*_10*X*_—coefficients obtained by regression analysis.

To determine the quantitative composition of the microstructure formed in the course of welding, several diagrams were constructed based on these equations, separately for each section of a welded joint. When constructing diagrams, austenitizing temperature and cooling rates are pre-set individually. The austenitizing temperature is equal to the maximum heating temperature of this area as far as the TEZ different parts are concerned. As for the weld, the *T_A_* is 1350 °C. The range of cooling rates was chosen based on maximum possible design instantaneous cooling rates according to the formulas obtained from the expression (3). Diagrams were constructed in the coordinates “% of structural components (*X*)—cooling rate (*W_cool_*)”. A criterion for determining the structural composition is the maximum instantaneous cooling rate obtained from calculations for a given part of a welded joint.

The experimental studies were conducted with regard to the specimens welded according to the modes shown in Table 2 and included the metallographic analysis of the macro- and microstructure of welded joints, determination of hardness, and mechanical properties in standard static tension tests. The metallographic analysis was conducted using the Altami MET 1T optical microscope and the VideoTest Metall image analysis software system. The surface of microslices was treated alternately with two reagents and multiple repolishing (the first reagent based on nitric acid, and the second one based on picric acid). Hardness was determined using the PMT-3 instrument and the Tukon 2500 Vickers hardness tester.

## 3. Results and Discussion

The results of experimental and calculation studies of the weld metal crystallization process in EBW (20Cr3MoWV steel) are presented in Figure 5, which also presents the weld macrostructure (a), the shape of a crystallizing part of the welding bath with crystallite axis projections (b), and the diagram of changes in the macrostructure shape and crystallization scheme (c). The shape and sizes of the crystallizing part were determined by the thermal model (1, 2); the crystallite axis projections were determined by the equations (4, 5); the diagrams of changes in macrostructure shape and crystallization scheme were based on the calculations of integral crystallization criteria (6–11).

The results show that the similar depth of fusion welding with combined beam oscillation trajectory leads to an increase in the sizes of the crystallizing part of the welding bath, in relation to welding without beam sweep and with x-shaped oscillations. At the same time, the weld shape in cross-section is close to that of the weld produced by welding with x-shaped oscillations and differs by its width. An increase in the sizes of the crystallizing part affects the crystallization process and the macrostructure formation; the width of a central zone with equiaxed grains increases, and there is no flat growth pattern for columnar crystallites.

The calculations of thermal cycles and instantaneous cooling rates were made for the following parts of welded joints:In the depth of a welded joint: for the upper—0.1*H*, middle—0.5*H* and root—0.9*H* parts;In the width of a welded joint: for the weld metal, for the overheating area, temperature—1350 °C, for the full recrystallisation area, temperature—1000 °C.

An example of calculated thermal cycles and instantaneous cooling rates for the TEZ overheating area is given in Figure 6.

The calculations show that the extension of a fusion zone in welding with beam oscillations along the combined trajectory contributes to the fact that the time of metal residence in its liquid state increases and the range of maximum instantaneous cooling rates for crystallized metal decreases. Thus, in the weld metal produced from welding without beam sweep, the maximum instantaneous cooling rate at the temperature of 1350 °C varies in the depth of fusion from 900 to 3600 °C/s; in the weld metal produced from welding with x-shaped beam oscillations, it varies from 460 to 2700 °C/s, and in the weld metal produced from welding with combined beam sweep—from 220 to 1250 °C/s. In the thermal effect zone, the time of metal residence increases above the critical *A*_3_ point (Figure 6), and the level of maximum instantaneous cooling rates, as well as their range, decreases as compared to welding without beam oscillations and with x-shaped oscillations. In the TEZ part heated up to the maximum temperature of 1350 °C, the range of instantaneous cooling rates is 610–2200 °C/s when welding by a static beam and 190–980 °C/s when welding with combined beam sweep. Maximum instantaneous cooling rates decrease in the TEZ parts heated up to lower temperatures, but more than the *A*_3_ point. In the area heated up to the maximum temperature of 1000 °C when welding by a static beam, their range is 200–750 °C/s, and when welding with combined beam sweep, their range is 85–480 °C/s.

A decrease in the difference between cooling rates in the depth of a welded joint creates conditions for reducing the degree of structural non-uniformity. To determine the quantitative composition of structural components in welded joints, a calculation method for analyzing the emerging microstructure of welded joints was used, and metallographic studies were conducted.

To calculate the volume fraction of formed structural components in welded joints, structural diagrams were constructed depending on the cooling rate (12–13) in the coordinates “% of structural components (*X*) —cooling rate (*W_cool_*)”. When constructing the diagrams, the following austenitizing temperatures were pre-set: for the weld metal and the TEZ overheating area, the *T_A_* = 1350 °C, for the TEZ full recrystallization area, the *T_A_* = 1100 °C. A percentage ratio of structural components is determined according to the range of maximum instantaneous cooling rates for a given area. An example of determining the volume fraction of structural components is presented in Figure 7.

According to the presented calculation results, a practically martensitic structure should be formed in the TEZ when welding by a static beam: in the upper part, the amount of bainite does not exceed 10%, while in the lower part of the weld, it does not exceed up to 2%. The application of x-shaped beam oscillations should lead to an increase in bainite in the upper part up to 10–12%, and in the root part up to 2–5%. When welding with combined beam oscillations, about 20% of bainite can be formed in the TEZ upper part and within 5–10% in the lower part.

As far as it is known, conventional methods of metallographic analysis do not always make it possible to clearly differentiate structural components close in morphological structure, such as lower bainite and martensite. The method of multiple polishing with reagent alternation results in detecting a relief on the surface of a slice, which creates a more comprehensive idea of the morphology of resulting structures, especially under large increases. Polarized light was additionally used to differentiate structural components in microstructure analysis. Polarized light reflects the morphology of formed structures and singles out a carbide phase (present in bainite). The carbide phase in polarized light is singled out in the form of rounded light inclusions with clear outlines. Consequently, the areas where intermediate (bainite) transformation takes place will be tinted with a luminous ring of light. Martensite in polarized light looks darker; these dark inclusions reflect its morphology—a package structure.

The quantitative assessment of structural components was carried out using the VideoTest-Metal image analysis software system. This software system singles out structural components according to their brightness range and determines the volume fraction of a highlighted phase. The analysis was conducted with regard to five fields of view. Figure 8 shows a microstructure, an example of separating phases and determining their volume fraction for the middle part of weld metal. The measurement results regarding the quantitative composition of weld metal are shown in Table 3, including errors and the composition calculated according to the above procedure for comparison.

The emergence of bainite in the weld metal structure and the thermal effect zone is of great importance for the mechanical properties of heat-resistant steels. When welding these steels without further heat treatment, the most optimal combination of mechanical properties will be in welded joints with mixed martensite and bainite structure.

The degree of non-uniformity in mechanical properties was assessed by the nature of variation in microhardness in the width of a welded joint for areas different in depth. Microhardness was measured in the upper, middle, and lower parts of welded joints using the Vickers method. Based on the obtained information, the average values of weld metal microhardness and root-mean-square deviation (RMSD) were determined as a measure of non-uniformity (Table 4). When welding by a static beam in the depth of a weld, there is an increase in RMSD of microhardness values, which indirectly indicates an increase in mechanical inhomogeneity. When welding with combined beam sweep and with x-shaped oscillations, the average value of weld metal microhardness decreases. RMSD values also decrease, i.e., it is arguable that the degree of mechanical inhomogeneity is lower here.

The specimens cut out in the cross-section of a welded joint were used for tension tests. Strength and flexibility properties were determined for the weld metal. The specimens were fractured at the weakest point of a welded joint. The results are given in Table 5.

The results show that welding with x-shaped beam oscillations and with combined sweep reduces the level of strength properties. Tensile strength *σ_B_* and constrained yield stress *σ*_0.2_ have similar values. At the same time, these values are much lower than *σ_T_* and *σ*_0.2_ of the weld metal produced from welding without beam oscillations. Similar values were also obtained for relative ultimate uniform elongation *δ_r_*. Metal flexibility margin was assessed by ultimate strength against yield strength (*σ*_0.2_/*σ_V_*), and if *σ*_0.2_/*σ_V_* = 1, metal flexibility margin was considered to be zero. As for medium-alloy steels, a ratio of yield strength against ultimate strength is permissible within the range of 0.7–0.8. Weld metal flexibility margin is slightly higher in case of welding with combined sweep.

Considering the foregoing, the proposed EBW method with dynamic beam positioning along the combined trajectory ensures a significant decrease in structural and mechanical inhomogeneity and brings the level of mechanical properties closer to that of the parent material. Consequently, this EBW method will be preferable for heat-resistant steel structures not subjected to further heat treatment.

## 4. Conclusions

The application of electron beam oscillations in welding makes it possible to change a welding pool shape and dimensions, as well as affect the crystallization and the formation of a primary structure. The emergence of bainite in the weld metal structure and the thermal effect zone is of great importance for the mechanical properties of heat-resistant steels. When welding these steels without further heat treatment, the most optimal combination of mechanical properties will be in welded joints with mixed martensite and bainite structure. The use of the trajectory of dynamic positioning of an electron beam with heating of edges of a welded joint in EBW leads to an increase in the width of a fusion zone.

At almost the same depth of fusion, welding with combined beam oscillation trajectory leads to an increase in the sizes of the crystallizing part of the welding bath, in relation to welding without beam sweep and with x-shaped oscillations. The weld shape in cross-section is close to that of the weld produced by welding with x-shaped oscillations and differs by its width. An increase in the sizes of the crystallizing part affects the crystallization process and the macrostructure formation: the width of a central zone with equiaxed grains increases and there is no flat growth pattern for columnar crystallites. The extension of a fusion zone increases the weld metal’s residence time and the thermal effect zone above the critical *A*_3_ point, increases cooling time, and considerably reduces instantaneous cooling rates and their range of change in depth respectively. This further reduces the degree of non-uniformity of macro- and microstructure in welded joints. As a result, welding of heat-resistant steels reduces the degree of non-uniformity of mechanical properties in the depth of welded joints, as well as reduces the level of hardening of a welded joint in relation to parent metal.

## Figures and Tables

**Figure 1 materials-13-02233-f001:**
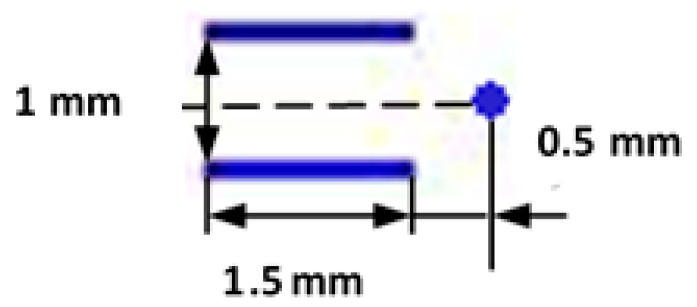
Trajectories of dynamic electron beam positioning.

**Figure 2 materials-13-02233-f002:**
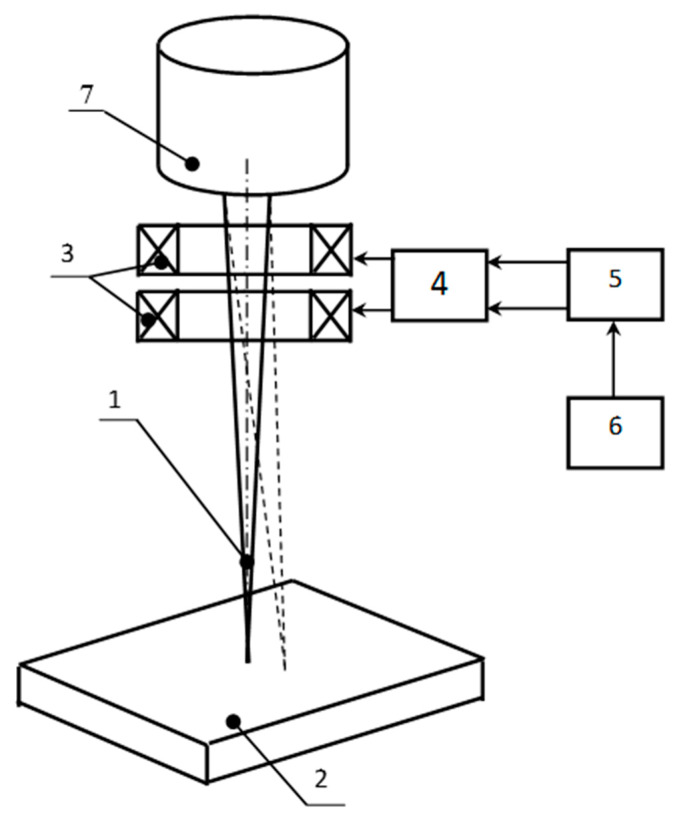
Scheme of the high-speed deflection system connection: 1—electron beam; 2—workpiece; 3—deflection coils; 4—broadband amplifier; 5—digital-to-analog interface; 6—computer; 7—electron beam gun.

**Figure 3 materials-13-02233-f003:**
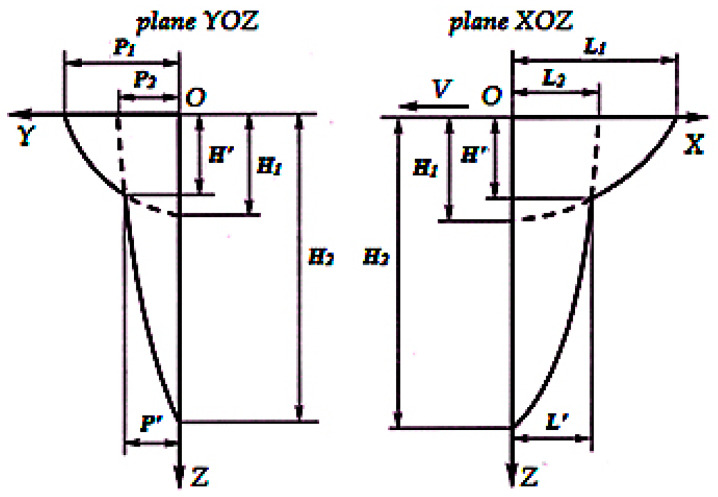
Schematic division of the crystallization front into components: *P*′, *H*′, and *L*′—intersection point coordinates for the two curves (inflection point); *P*_2_, *H*_1_, and *L*_2_—values at which the curves intersect relevant coordinate axes; *P*_1_—corresponds to the value of the weld half-width, *H*_2_—corresponds to the weld depth; *L*_1_—numerically equal to the length of the welding bath on the surface.

**Figure 4 materials-13-02233-f004:**
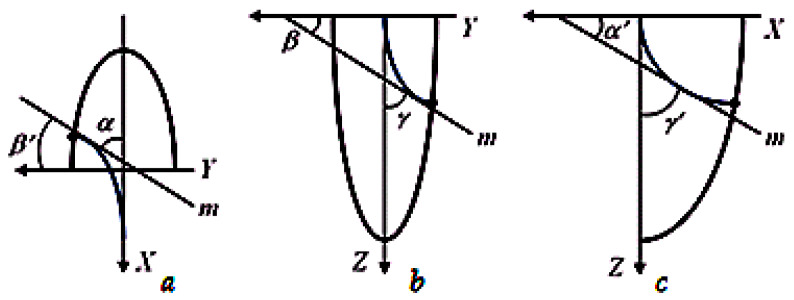
Direction angles of the tangents (m) to the crystallite axis: (**a**) *α* and *β*′ in the *XOY* plane, (**b**) *β* and *γ* in the *YOZ* plane, (**c**) *α*′ and *γ*′ in the *XOZ* plane.

**Figure 5 materials-13-02233-f005:**
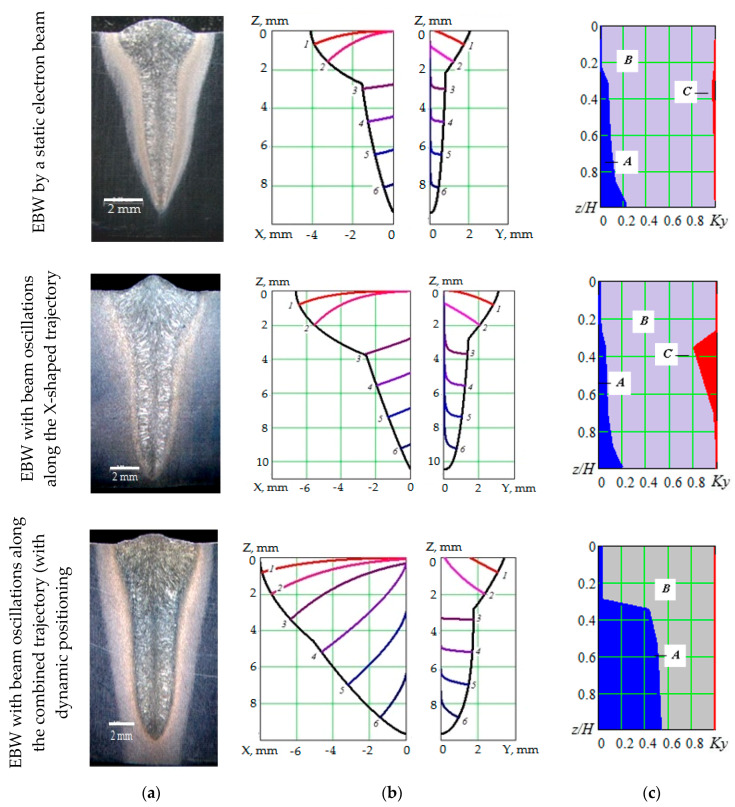
Weld macrostructure (**a**), shape of the crystallizing part of the welding bath with crystallite axis projections in the XOZ and YOZ coordinate planes (**b**), and diagrams of changes in macrostructure shape and crystallization pattern in provisional coordinates in the *z/H* depth and in the *Ky* = *y/y*_0_ (**c**) weld width: (A—equiaxial structure, B—spatial growth of columnar crystallites, C—flat growth of columnar crystallites).

**Figure 6 materials-13-02233-f006:**
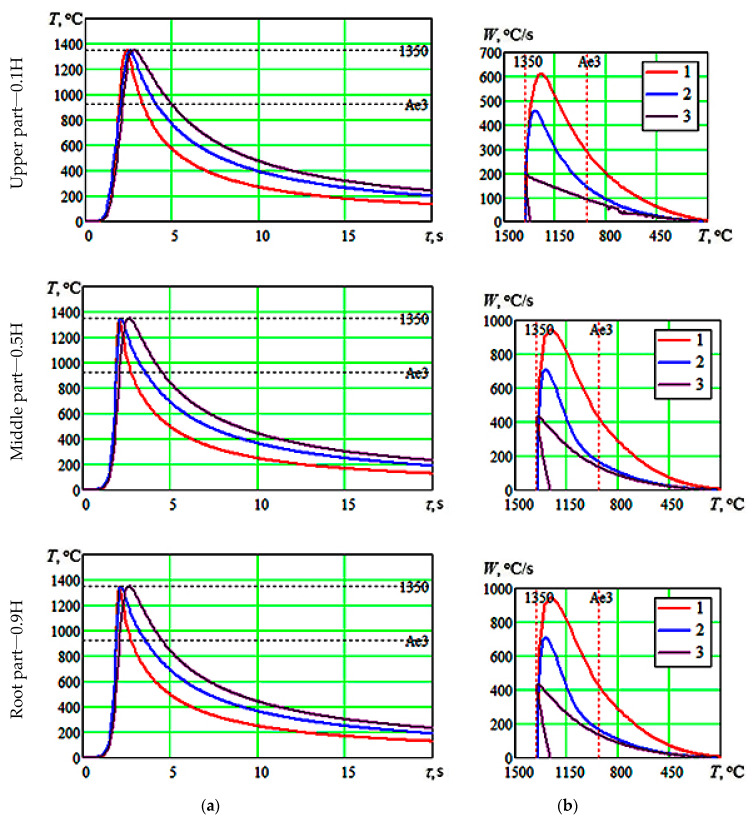
Calculated thermal cycles (**a**) and instantaneous cooling rates (**b**) in the TEZ overheating area (maximum heating temperature—1350 °C) for different weld depths: 1—welding without beam sweep; 2—welding with x-shaped beam oscillations; 3—welding with combined beam sweep.

**Figure 7 materials-13-02233-f007:**
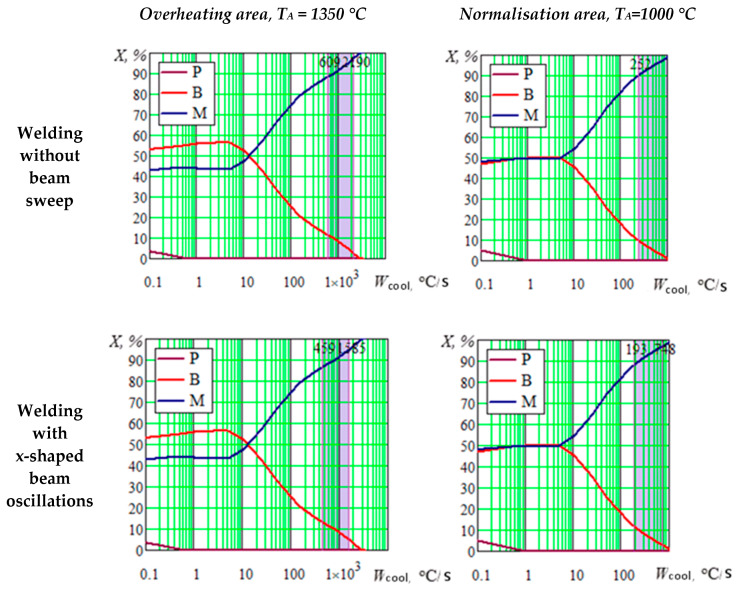
Structural diagrams for the TEZ two parts (a welded joint from the 20H3MVF steel): P—perlite, B—bainite, M—martensite (the diagrams show the range of instantaneous cooling rates in the depth of welded joints).

**Figure 8 materials-13-02233-f008:**
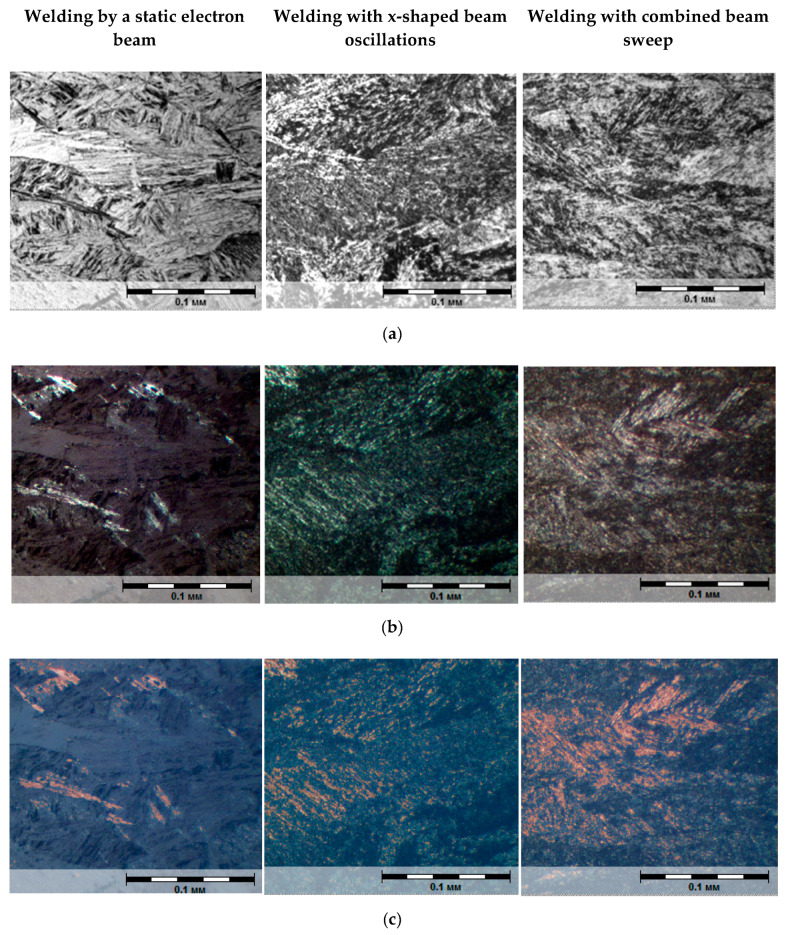
Weld metal microstructure in the middle (**a**) part and an example of determining a percentage ratio of structural components: polarized light (**b**), separating structural components in the VideoTest-Metal program (**c**), ×500.

**Table 1 materials-13-02233-t001:** Chemical composition of the 20Cr3MoWV steel, wt %.

Fe	C	Mn	Si	Cr	Mo	V	W	Ni	P	S	Cu
								Not More than			
Base	0.15–0.23	0.25–0.5	0.17–0.37	2.8–3.3	0.35–0.55	0.6–0.85	0.6–0.85	0.03	0.03	0.025	0.025

**Table 2 materials-13-02233-t002:** EBW parameters for the 20Cr3MoWV steel.

Type of EBW	Beam Power, W	Welding Speed Vweld, mm/s	Sweep Parameters	
			Frequency, Hz	Amplitude, mm
Without beam sweep	2000	5	-	-
With X-shaped oscillations	4000	5	800	2
With a sweep along the trajectory presented in Figure 1	4000	5	1000	-

**Table 3 materials-13-02233-t003:** Structural composition of weld metal obtained experimentally and by calculation.

Type of Electron Beam Sweep	Distance in the Depth pf Joint	Experimental Observation	Calculation Based on Regression Dependencies
Without beam sweep	0.1H	7.3% V, 92.7% M (ε − 5.7%)	9% V, 91% M
	0.5H	4.8% V, 95.2% M (ε − 3.11%)	4% V, 96% M
	0.9H	2.9% V, 97.1% M (ε − 4.35%)	0% V, 100% M
With X-shaped oscillations	0.1H	18.8% V, 8.2% M (ε − 2.8%)	14% V, 86% M
	0.5H	14% V, 86% M (ε − 2.5%)	8.5% V, 91.5% M
	0.9H	9.3% V, 90.7% M (ε − 2.9%)	2% V, 98% M
Three-beam welding	0.1H	23% V, 77% M (ε − 4%)	19% V, 81% M
	0.5H	20% V, 80% M (ε − 4.2%)	14% V, 86% M
	0.9H	13% V, 87% M (ε − 9%)	7% V, 93% M

**Table 4 materials-13-02233-t004:** Changes in weld metal hardness for different types of EBW.

Part	Average Hardness Values	Welding by a Static Electron Beam	Welding with x-Shaped Beam Oscillations	Welding with Combined Beam Sweep
Weld upper part	Aver. HV 0.1	481	404	418
	RMSD	94	60	66
Weld middle part	Aver. HV 0.1	479	394	413
	RMSD	127	80	67
Weld lower part	Aver. HV 0.1	488	410	394
	RMSD	136	88	64

**Table 5 materials-13-02233-t005:** Mechanical properties of welded joints produced with different types of EBW.

Mechanical Properties	Type of EBW			Parent Metal
	Welding by a Static Electron Beam	Welding with x-Shaped Beam Oscillations	Welding with Combined Beam Sweep	
*σ_V_*, MPa	1535	1333	1301	822
*σ*_0.2_, MPa	1397	1193	1159	75
*σ*_0.2_/*σ_V_*	0.91	0.894	0.882	0.75
*δ_r_*, %	4.1	4.43	4.49	8.84

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
