# Peer review of "Application of Dynamic Beam Positioning for Creating Specified Structures and Properties of Welded Joints in Electron-Beam Welding"

_materials, 2020, doi:10.3390/ma13102233_

Round 1
Reviewer 1 Report
Dear Authors,
The issue of your paper is very interesting, but the way selected to describe it is not satisfactory. Thus, my first approach was to reject the paper, but I would like to give you an extra opportunity to improve it, as follows:
- The title is not attractive. Please try to change "Formation of a welded joint...".
- The Introduction is clearly short and ill-designed. The number of papers referred is very low and the quality of the sources of information is extremely low. Most of the journals have not Impact Factor from Clarivate Analytics.
- References [8] and [9] are not referred in the Introduction. But reference [10] is. Thus, please reorganize the references.
- At the final of the Introduction, please provide a framework of your paper.
- In the Methods, please provide information about the equipment and parameters used to carry out SEM observations and Tensile Testing.
- Please provide information about the CONVECTION effect in the welding pool, regarding your first paragraph of section 2.
- In Figure 1, please don't use capital letters to express [mm].
- After Figure 1, are you talking about Specific Power?
- Please explain the meaning of the following sentence in page 2: "The distance between 66 the electron beam exposure zones was to have one general fusion channel formed by the electron 67 beam in the metal.".
- Please provide information and calculations about the heat passed to the joint, the peak temperature and the cooling rate.
- Regarding the last paragraph of page 2, please provide a diagram showing how the different devices are connected.
- Could you provide information about the diameter of the beam when it reaches the part to be welded?
- Regarding Figures 2, 3 and 4, maybe (c) and (d) are exchanged.
These images need to be explained, as well pointed out the software and calculations made to reach these graphs. This information should be put in the Methods section. - The graphs of Figure 5 don't show the units in the X scale. Moreover, some scales seem cut. Thus, it is difficult to understand the differences and whar they intend to describe. You say that "(the diagrams show the range of instantaneous cooling rates in the depth of welded joints)". Why, if the upper row is extremely similar to the 2nd row? What means the shadow in the right?
- The paragraph after Figure 5 intends to describe what we are seeing in Figure 5, but the values don't agree. You must explain.
- In Table 2, what means "Indentation Tension" as method of determination?
- There is no discussion about your results. Moreover, they are not commented...
- The Conclusions need to be supported by the Results, and it doesn't happen. Where do you refer the shape and dimensions of the welding pool in the work (Results)? What kind of analysis is made to the crystals obtained ou microstructure?
You need to deeply revise the manuscript because, as it is, it is not useful for the readers and for the Science.
Kind regards.
Reviewer
Author Response
"Please see the attachment."

Reviewer 2 Report
Dear authors, thank you for the interesting contribution.
The paper concerning experimental and calculation studies on the effect of electron-beam welding on the formation of structures and properties of 20Cr3MoWV steel welded joints.
I have some answers/suggestions:
- The manuscript introduction may be implemented, but it provides sufficient backgrounds.
- Materials and methods section must be implemented by adding details about the mechanical tests performed; I realized they were made only at the end of the discussion paragraph.
- All figures and tables captions must be implemented because the graphs lack in the unit of measurement; in Fig.4-c the point C was not indicated but was discussed in its caption.
- Fig.2, 3 and 4 lack in the indication of the EBW process adopted. Moreover, all the images b and c must be further explained in the text to better understand the EBW effect and they lack in the name of the axis. Z, H, Ky parameters must be explained.
- May you write mm instead of MM in Fig.1?
- The comma was used as both the decimal point and as an indicator of thousands (lines 59, 60 as decimal point, lines 70, 71, as thousands indicator). Besides, in the figures was adopted the point. Could you replace the comma?
- Figure 5 is not very clear. They look identical. X% as y-axis represents the phase percentage? Some numerical parameters were indicated on the diagrams and they are not very readable.
- In line 105 (tab. 2) are not clear the total measurements/ tests performed. How many samples were used? How many indentations were performed?
- Why did you not show the state of the art values for the attended mechanical properties? Are the results coherent with the expectative or are lower/ higher? In the conclusions, you said "welding of heat resistant steels reduces the degree of non-uniformity of mechanical properties in the depth of welded joints" but was not shown a standard deviation of the mechanical values to corroborate this argument. What were you mean?
In general, the discussion of the figures may be enhanced, figures must be implemented by extended-axis-name (no unknown abbreviation) and captions must be improved; only for these reasons, I suggest major revisions.
Best regards
Author Response
"Please see the attachment."

Round 2
Reviewer 1 Report
Dear Authors,
Congratulations on your work, which was considerably improved.
However, some final adjustments need to be done before it can be accepted for publication, namely:
- The paper needs to be proof-read, because there are some basic errors. An example: Page 1, Line 39: "It is know that EBW is use in...", when you should write "It is known that EBW is used in...". As this, tehere are several situations with wrong conjugations of the verbs and grammatical errors. Thus, please improve the text or order Editing Services. or ask a English Native Speaker to clean the text.
- There are some mistakes in formatting your paper, such as the wrong use of non-capitalized letters and so on (last line o Figure 3), the wrong use of dots (line 139 -> [16,17.]).
After these adjustments, your paper can be accepted for publication.
Please pay attention to all details.
Kind regards,
Reviewer
Author Response
Dear Assistant Editor MDPI Materials Journal
Ms. Karena Tang,
Dear Reviewers
Please accept our sincerest thanks for your kind feedback
We used a professional English editing service (https://www.manuscriptedit.com/) to improve the quality of the English language and remove typos.
The" Page 1, Line 39: "It is know that EBW is use in...", when you should write "It is known that EBW is used in...". " and "...the wrong use of dots (line 139 -> [16,17.])." have been fixed too
Reviewer 2 Report
Dear authors,
thank you for revising your manuscript.
You perform extended revisions that increase a lot the total impact of your interesting work.
For this reason, I'll suggest acceptance in the present form.
Kind regards
Author Response
Dear Assistant Editor MDPI Materials Journal
Ms. Karena Tang,
Dear Reviewers
Please accept our sincerest thanks for your kind feedback
We used a professional English editing service (https://www.manuscriptedit.com/) to improve the quality of the English language and remove typos.